Downregulation of the enhancer of zeste homolog 1 transcriptional factor predicts poor prognosis of triple-negative breast cancer patients

Peng Wei 1
Tang Wei 2
Li Jian-Di 2
He Rong-Quan 1
Luo Jia-Yuan 3
Chen Zu-Xuan 4
Zeng Jiang-Hui 5
Hu Xiao-Hua 1
Zhong Jin-Cai 1
Li Yang 1
Ma Fu-Chao 1
Xie Tian-Yi 6
Huang Su-Ning 7 huangsuning@stu.gxmu.edu.cn
Ge Lian-Ying 8 gxgly@hotmail.com
1 Department of Medical Oncology, The First Affiliated Hospital of Guangxi Medical University , Nanning, Guangxi , China
2 Department of Breast Surgery, Guangxi Medical University Cancer Hospital , Nanning, Guangxi , China
3 Department of Pathology, First Affiliated Hospital of Guangxi Medical University , Nanning, Guangxi , China
4 Department of Medical Oncology, The Second Affiliated Hospital of Guangxi Medical University , Nanning, Guangxi , China
5 Department of Clinical Laboratory, The Third Affiliated Hospital of Guangxi Medical University/Nanning Second People’s Hospital , Nanning, Guangxi , China
6 Department of Traditional Chinese Medicine, The First Affiliated Hospital of Guangxi Medical University , Nanning, Guangxi , China
7 Department of Radiotherapy, Guangxi Medical University Cancer Hospital , Nanning, Guangxi , China
8 Department of Endoscopy, Guangxi Medical University Cancer Hospital , Nanning, Guangxi , China
Gould Gwyn
Electronic publication date: 2022 Jul 12
Publication date: 2022
Volume: 10
Electronic Location ID: e13708
Received 2022 Mar 2; Accepted 2022 Jun 19
Copyright: © 2022 Peng et al.
Copyright year: 2022
Copyright holder: Peng et al.
License: This is an open access article distributed under the terms of the Creative Commons Attribution License, which permits unrestricted use, distribution, reproduction and adaptation in any medium and for any purpose provided that it is properly attributed. For attribution, the original author(s), title, publication source (PeerJ) and either DOI or URL of the article must be cited.
License URL: https://creativecommons.org/licenses/by/4.0/

Keywords: Triple-negative breast cancer, EZH1, Transcriptional regulation, Molecular approaches

Funding: National Natural Science Foundation of China NSFC82060309 Natural Science Foundation of Guangxi, China 2022GXNSFAA Key R&D Plan of Science and Technology Plan Project in Qingxiu District, Nanning City 2020020 Guangxi Zhuang Autonomous Region Health Commission Self-Financed Scientific Research Project Z2014245 This study was funded by the National Natural Science Foundation of China (NSFC82060309), the Natural Science Foundation of Guangxi, China (2022GXNSFAA), the Key R&D Plan of Science and Technology Plan Project in Qingxiu District, Nanning City (2020020), and the Guangxi Zhuang Autonomous Region Health Commission Self-Financed Scientific Research Project (Z2014245). The funders had no role in study design, data collection and analysis, decision to publish, or preparation of the manuscript.

==============================
Background

Triple-negative breast cancer (TNBC) is the most malignant subtype of breast cancer and lacks effective biomarkers. This study seeks to unravel the expression status and the prospective transcriptional mechanisms of EZH1/EZH2 in TNBC tissue samples. Moreover, another objective of this study is to reveal the prognostic molecular signatures for risk stratification in TNBC patients.

Methods

To determine the expression status of EZH1/EZH2 in TNBC tissue samples, microarray analysis and immunohistochemistry were performed on in house breast cancer tissue samples. External mRNA expression matrices were used to verify its expression patterns. Furthermore, the prospective transcriptional mechanisms of EZH1/EZH2 in TNBC were explored by performing differential expression analysis, co-expression analysis, and chromatin immunoprecipitation sequencing analysis. Kaplan–Meier survival analysis and univariate Cox regression analysis were utilized to detect the prognostic molecular signatures in TNBC patients. Nomogram and time-dependent receiver operating characteristic curves were plotted to predict the risk stratification ability of the prognostic-signatures-based Cox model.

Results

In-house TMAs (66 TNBC vs. 106 non-TNBC) and external gene microarrays, as well as RNA-seq datasets (1,135 TNBC vs. 6,198 non-TNBC) results, confirmed the downregulation of EZH1 at both the protein and mRNA levels (SMD = −0.59 [−0.80, −0.37]), as is opposite to that of EZH2 (SMD = 0.74 [0.40, 1.08]). The upregulated transcriptional target genes of EZH1 were significantly aggregated in the cell cycle pathway, where CCNA2, CCNB1, MAD2L1, and PKMYT1 were determined as key transcriptional targets. Additionally, the downregulated transcriptional targets of EZH2 were enriched in response to the hormone, where ESR1 was identified as the hub gene. The six-signature-based prognostic model produced an impressive performance in this study, with a training AUC of 0.753, 0.981, and 0.977 at 3-, 5-, and 10-year survival probability, respectively.

Conclusion

EZH1 downregulation may be a key modulator in the progression of TNBC through negative transcriptional regulation by targeting CCNA2, CCNB1, MAD2L1, and PKMYT1.

Introduction

Breast cancer (BC) remains one of the most common malignant tumors in women (Sung et al., 2021). Due to the huge heterogeneity of BC, the therapeutic responsiveness varied from patients to patients (Ding, Chen & Shen, 2020). As early as 2013, the molecular classification of BC has been endorsed by the St Gallen Consensus Conference (Goldhirsch et al., 2013; Tsang & Tse, 2020). According to immunohistochemistry (IHC) classification, BC can be roughly divided into the luminal A subtype, luminal B subtype, human epidermal growth factor receptor 2 (HER2) overexpression subtype, and triple-negative breast cancer subtype (TNBC) (Stovgaard et al., 2019). TNBC refers to a subtype of BC that displays negative expression of the estrogen receptor (ER), progesterone receptor (PR), and HER2 (Hua, White & Zhou, 2021). In the study of 78,708 primary TNBC patients from 2010 to 2014, the 3-year survival rate of non-Hispanic black women is 79.4%; and the 3-year survival rate is only 14.4% for those Asian women with stage IV TNBC (Wang et al., 2021a). Due to a lack of effective targeted therapies, TNBC—accounting for nearly 15% of all BCs—has the worst prognosis compared with any other BC molecular subtypes (Gupta et al., 2020; Hosonaga, Saya & Arima, 2020). Although immune checkpoint blockages may have seen a promising effect in treating advanced TNBC patients (Gradishar et al., 2020), there seems to be more emerging challenges, such as neuropathy, de novo or acquired treatment resistance, and so on (Vagia, Mahalingam & Cristofanilli, 2020). With the rapid development of next-generation sequencing and single-cell RNA-sequencing (RNA-seq) technologies, it is possible to evaluate the prognosis and therapies of TNBC patients using a personalized medicine approach (Garrido-Castro, Lin & Polyak, 2019; Jiang et al., 2019; Verret et al., 2020). Therefore, more efforts deserve to be made to study novel prognosticators and interventional targets for treating TNBC patients.

Recently, enhancer of zeste homolog 1 (EZH1) transcriptional factor (TF), a member of the polycomb group (PcG) protein family, has been reported to participate in the proliferation and metastasis of tumor cells (Dou et al., 2019; Li et al., 2019; Yamagishi et al., 2019). Normally, EZH1 is predominantly expressed in the nucleoplasm and is a component of polycomb repressive complex 2, which is essential for the pluripotency of embryonic stem cells (Glancy, Ciferri & Bracken, 2021; Shen et al., 2008). Functionally, both EZH1 and EZH2 are involved in the trimethylation of lysine at position 27 of histone H3 (H3K27me3), which are responsible for the transcriptional repression of downstream target genes (Meng et al., 2020; Rizq et al., 2017). Therefore, EZH1/EZH2 have been used to design drugs for hampering tumor growth and restoring tumor suppressor transcription (Damele et al., 2021; Healy et al., 2019). More intriguingly, in case of EZH2 dysfunction, EZH1 can serve as backup enzyme of EZH2 (Kusakabe et al., 2021; Shen et al., 2008). EZH1 can not only partially compensate the absence of EZH2 in constituting H3K27me1, but also may exert other non-redundant roles in epigenetic establishment of cell fates (Shen et al., 2008). In BC tissue and cell lines (i.e., MCF-7 and SKBR3), EZH1 has been demonstrated to be upregulated and is negatively regulated by microRNA-765 (Zeng, Yang & Wu, 2019). Nevertheless, another study elaborated that EZH1 was downregulated in SUM159 BC cells (Liu et al., 2012), which could be induced by miRNA-93 expression; miRNA-93 has proved to be upregulated in the MDA-MB-231 cell line and ER- or PR-negative BC patients (Kolacinska et al., 2014; Li et al., 2017). In this context, the expression patterns of EZH1 in BC are controversial. Furthermore, the expression levels and transcriptional regulatory mechanisms of EZH1 in TNBC are not known. Therefore, more experimental evidence must be provided to clarify the potential role of EZH1 in TNBC.

Furthermore, non-coding RNAs, such as long non-coding RNAs (lncRNA) and microRNAs (miRNA), also play critical roles in the tumorigenesis of TNBC (Wang et al., 2021b; Zhang, Guan & Tang, 2021). For instance, it was reported that lncRNA titin‑antisense RNA1 deletion could attenuate the cell viability and invasion ability of TNBC cells (Sun et al., 2021). Also, lncRNA was involved in the cis-regulation and trans-regulation of genes by interacting with TF and histone-modifying enzymes (Wong, Tsang & Ng, 2018). However, little was known on the potential interplay between EZH1 and non-coding RNAs.

Based on the current situation, this study seeks to unravel the expression status and the prospective transcriptional mechanisms of EZH1/EZH2 in TNBC tissue. Moreover, another objective of this study is to reveal the prognostic molecular signatures for risk stratification in TNBC patients.

Materials and Methods

Breast cancer (BC) tissue microarrays

A total of three tissue microarrays (TMAs), namely HPreD140Su06 (140 tissue cores), HPreD077Su01 (77 tissue cores), and HPreD050Bc01 (50 tissue cores), were purchased from Shanghai Xinchao Biotechnology Co., Ltd. The quality of the TMA samples was strictly controlled using the following criteria: (I) normal breast tissue or BC with histological classification were pathologically diagnosed using conventional hematoxylin and eosin (HE) staining sections; (II) sufficient clinicopathological information of BC patients was provided, including IHC results of ER, PR, HER2, HER2 fluorescence in situ hybridization, and so on; and (III) the patient had not received any anti-tumor therapy before surgery. TMA samples were excluded: (I) BC samples came from male patients; (II) there were insufficient IHC results for classifying the molecular subtypes of BC; (III) specimens were pathologically diagnosed with a special type of BC, such as medullary carcinoma, mucinous carcinoma, and so on; and (IV) fragmentation was found during IHC staining, thus making it difficult to display the protein expression results of EZH1.

Immunohistochemistry (IHC) staining

A two-step IHC staining method was used in this study. Tissue chips were baked at 65 °C overnight, and were dewaxed for 5 min, then Citrate buffer was used for antigen retrieval under high pressure for 2 min and 30 s. Endogenous peroxidase activity was blocked using 3%H2O2. After preliminary experiments, it was determined that the best dilution concentration of the primary rabbit polyclonal antibody anti-EZH1 was 1:100 (https://www.abcam.com/ezh1-antibody-ab64850.html). The Zhongshan Golden Bridge PV-6000 was selected as the secondary antibody. Then, a DAB Kit (20×) (DAB-0031) was used for color rendering. When interpreting the IHC staining results, a semi-quantitative score method was used, which was recorded in Table S1. The raw data of in-house IHC result was recorded in Table S2.

mRNA-expression matrices of BC tissue

mRNA datasets of BC tissue and adjacent normal tissue were accessed from ArrayExpress, cBioPortal, Gene Expression Omnibus (GEO), The Cancer Genome Atlas (TCGA), and the Sequence Read Archive. The keywords searched were (breast cancer OR breast carcinoma) AND “Homo sapiens.” The inclusion standards were as follows: (I) the study type should be an expression profiling by array or by high throughput sequencing, and (II) the clinicopathological parameters of each BC patient should be provided for group division. Duplicate samples and treated patients were excluded. The included mRNA matrices were grouped by GEO platforms and merged into platform datasets. The known batch effect produced in the process of dataset integration was eliminated using the ComBat function with an empirical Bayes framework. The NormalizeBetweenArrays function was used to perform data normalization to certify that the expression values had similar distributions across a platform. The batch-effect-eliminated and normalized platform matrices were matched to the clinic-pathological parameters of each individual. Then the mRNA-expression data of TNBC and non-TNBC tissue were abstracted from batch-effect-eliminated and normalized platform matrices for differential expression and co-expression analysis.

Cancer cell line encyclopedia and dependency map

Cancer Cell Line Encyclopedia (CCLE) sequencing data were downloaded to validate the expression status of EZH1/EZH2 in BC cell lines (https://depmap.org/portal/gene/EZH1?tab=characterization; https://depmap.org/portal/gene/EZH2?tab=characterization). We also explored the clustered regularly interspaced short palindromic repeats (CRISPR) screen data of EZH1/EZH2 in BC cell lines using the Dependency Map (DepMap). Gene effect scores (CERES) were used to evaluate the biological functions of EZH1/EZH2 in BC cell lines. Negative CERES may reflect that the growth of BC cancer cells could be inhibited by the knockdown of EZH1 or EZH2.

Survival data from triple-negative breast cancer subtype (TNBC) cohorts

To evaluate the comprehensive prognostic value of EZH1/EZH2, the overall survival (OS), distal metastasis-free survival (DMFS), and prognosis-free survival (PFS) data of the BC patients were gathered from the Kaplan Meier plotter, which included the prognostic data from 48 GEO gene microarrays and two ArrayExpress datasets (Győrffy, 2021). TNBC patients were subsequently filtered to evaluate the prognostic value of EZH1/EZH2 in TNBC patients. TCGA, METABRIC, and GSE25307 were selected to calculate the pooled hazard ratio (HR) of EZH1, all of which contained no less than 100 TNBC patients.

Prognostic molecular signatures in TNBC patients

Kaplan–Meier survival analysis was performed to screen for potential prognosis indicators in the TCGA TNBC cohort. Univariate Cox regression analysis was used to compute HRs from three large cohorts of TNBC patients independently (i.e., TCGA, METABRIC, and GSE25307). The HRs of each screened prognosis indicator were pooled to determine the prognostic molecular signatures in TNBC patients (risk factor: pooled HR > 1, P < 0.05; protective factor: pooled HR < 1, P < 0.05). Risk score of each patient was computed by using the following equations:

PooledHR=ecoefficient (e stands for natural logarithm) Riskscore=∑n=i⁡(coefficienti∗expression_valuei)

Nomogram and time-dependent receiver operating characteristic curves were plotted to predict the risk stratification ability of the prognostic-signatures-based Cox model.

Consensus clustering analysis of TNBC patients

Based on the identified prognostic signatures, TNBC patients in the TCGA cohort were subdivided using a method of consensus clustering. Delta area plot and cumulative distribution function (CDF) plot were utilized to select the optimal k value. The prognostic statuses of the different TNBC subtypes were compared.

Single-sample gene-set enrichment analysis

The immune microenvironments of different TNBC subtypes were portrayed by conducting a single-sample gene-set enrichment analysis (ssGSEA). The inputted immune metagenes were the same as previously demonstrated (Charoentong et al., 2017). Pearson’s correlation coefficients (PCCs) between EZH1 expression and immune infiltrations were calculated.

Somatic-mutation analysis of TNBC patients

The mutation landscapes of EZH1 and EZH2 were explored from TCGA and Metabric using cBioPortal. The somatic mutations of different TNBC subtypes were evaluated by reanalyzing Mutation Annotation Format (MAF) files from TCGA. Somatic interactions, especially mutation co-occurrence or exclusiveness, were studied using a paired Fisher’s exact test.

Differential expression analysis

Differently expressed genes (DEGs) were determined by pooling the standardized mean difference (SMD) value using the mRNA-expression data of TNBC patients, as is aforementioned in the “mRNA-expression matrices of BC tissue” section, compared with non-TNBC patients (Upregulated DEGs: SMD > 0, P < 0.05; Downregulated DEGs: SMD < 0, P < 0.05). All gene symbols containing 61,521 entries were used to filter DEGs in TNBC using a server.

Co-expression analysis

EZH1 or EZH2 co-expressed genes (CEGs) were determined by calculating PCCs using the mRNA-expression data of TNBC and non-TNBC tissue, which have been aforementioned in the “mRNA-expression matrices of BC tissue” section. CEGs were summarized from each platform matrix and were counted, which should be ≥3 (Positive CEGs: PCCs ≥ 0.3, P < 0.05; Negative CEGs: PCCs ≤ −0.3, P < 0.05).

Chromatin immunoprecipitation sequencing data analysis

Chromatin immunoprecipitation sequencing (ChIP-seq) data of EZH1 and EZH2 transcriptional target were queried and downloaded from the Cistrome database, one of the largest TF databases. Putative transcriptional targets of EZH1 and EZH2 were filtered by setting scores at ≥1. BigWig files were downloaded for peak identification. Integrative Genomics Viewer v2.8.13 software was utilized to visualize the peaks, where HG38 was selected as the human reference genome.

Prospective signaling pathways in TNBC patients

Gene ontology (GO), Kyoto Encyclopedia of Genes and Genomes (KEGG), and disease ontology (DO) annotations were conducted after intersecting upregulated DEGs, negative CEGs, and the predicted transcriptional targets. Downregulated DEGs, positive CEGs, and predicted transcriptional targets were also intersected and functionally annotated in the same ways. The mapped GO terms were clustered to identify the most enriched entries. Protein-protein interaction network was constructed to determine the hub gene in the pathway.

Potential TF-miRNA-lncRNA regulatory axis in TNBC patients

BC RNA-seq expression data in the format of fragments per kilobase of transcript per million reads mapped (FPKM) were downloaded from TCGA. FPKM data were transformed into transcripts per million (TPM) data using the following algorithm: TPM=exp(log(FPKM)−log(sum(FPKM))+log(1e6))

lncRNA TPM expression data were extracted from the BC TCGA matrix. Mature miRNA expression data in the format of reads of exon models per million mapped reads (RPM) were prepared for TNBC and non-TNBC specimens in TCGA. Differentially expressed miRNAs (DEmiRNAs) and differentially expressed lncRNAs (DElncRNAs) were calculated from TNBC tissue in comparison with non-TNBC tissue by using log2(TPM+1) data (log2|foldchange| > 1, P < 0.05). Putative binding miRNA targets of EZH1 were predicted by miRTarBase and TarBase V.8, which were intersected by DEmiRNAs. The upstream lncRNAs were forecasted by the encyclopedia of RNA interactomes and LncBase Predicted v.2 (score ≥0.8), which were intersected by DElncRNAs.

Drug repurposing by in-silico analysis

The connectivity map (Cmap) was utilized to identify novel therapeutic small molecules for TNBC patients that targeted the most enriched DEGs. The predicted therapeutic agents for TNBC were strictly limited by using the following criteria: (I) the forecasted drugs should be documented in the DrugBank, or (II) there should be experimental evidence to verify the anti-cancer effect of the forecasted drugs. The predicted small molecules were ranked by a normalized connectivity score, and the first-ranked molecule was used to dock with the transcriptional targets of EZH1. The binding energy threshold was −5 kcal/mol, and the lower binding energy, the more stable the molecule.

Statistical analysis

When comparing the expression levels of EZH1 or EZH2 between the two groups, an unpaired two-sample t-test was used. The total number (N), mean (M), and standard deviation (SD) of EZH1 and EZH2 expression values were calculated in TNBC and non-TNBC tissue. SMD was selected to appraise the overall expression levels of EZH1 and EZH2, where SMD > 0 or SMD < 0 implied overexpression and low expression, respectively. If the 95% confidence interval of SMD has no overlap with the zero, the SMD value will be significant. For the calculation of summary SMD and HR, a fixed effect model was selected when I2 < 50%. Provided that I2 was ≥50%, a randomized effect model should be selected. Begg’s test and Egger’s test were utilized to detect publication bias, where insignificant publication bias reflected the stability of the SMD result (Coelho-Júnior, Trichopoulou & Panza, 2021). We calculated the true positive, false positive, false negative, and true negative rates of EZH1 or EZH2 in TNBC compared with non-TNBC tissue, which was subsequently utilized for drawing summary receiver operating characteristic (SROC) curve. Area under the SROC curve (AUC) was selected to determine the discriminatory ability of EZH1 and EZH2 in TNBC and normal breast tissue, as well as intrinsic subtypes. AUC between 0.70–0.90 indicated a moderate discriminatory ability. A log rank test was performed in Kaplan–Meier (KM) survival analysis. P < 0.05 signified significance.

Results

EZH1 was downregulated in TNBC tissue

Previous studies reported controversial expression levels of EZH1 in BC, so the authors explored the expression pattern of EZH1 in TNBC tissue. Herein, a total of 264 tissue samples containing 177 BC and 87 normal breast tissue samples were eligible for evaluating the protein expression levels of EZH1. All samples had HE stain images to prove the pathological diagnosis (Figs. S1A–S1D). The baseline information of the in-house BC patients and the preliminary statistics of the external mRNA-expression matrices are presented in Tables 1 and 2, respectively. Compared with the non-TNBC tissue, both the EZH1 protein and mRNA-expression levels were significantly decreased in the TNBC tissue (Figs. 1A and 1B). Additionally, with an AUC value between 0.70 and 0.90, the ROC curves implied that there was a moderate discriminatory ability of EZH1 in differentiating between TNBC and non-TNBC tissue, which could be demonstrated by both the quantitative results of in-house IHC (Fig. 1C) and external gene microarrays and RNA-seq (Figs. 1D–1F). The IHC staining results of EZH1 in normal breast, TNBC, and non-TNBC tissue are displayed in Figs. 2–4, respectively, where EZH1 antibody was strongly stained in the nucleus of normal breast epithelial cells. Given the significantly downregulated expression level of EZH1 protein in TNBC, the authors subsequently analyzed the downregulation level of EZH1 in BC subgroups, and it was found that EZH1 protein was lowly expressed in TNBC tissue compared with normal breast, HER2-positive, or luminal BC tissue (Fig. S2).

Table 1 Baseline information of in-house breast cancer (BC) patients.

Characteristics	Case	EZH1 staining score	df	P value	
Age (years)	175	0.776	
≤50	65	3.91 ± 2.213			
>50	112	4.00 ± 1.991			
Histological grade	172	0.002	
II	108	4.24 ± 2.004			
II–III	43	4.16 ± 2.159			
III	22	2.59 ± 1.709			
Vascular cancer thrombus or nerve invasion	174	0.047	
Yes	35	3.34 ± 2.014			
No	141	4.12 ± 2.068			
Lymph node metastasis	175	0.131	
Yes	79	4.23 ± 2.025			
No	98	3.76 ± 2.091			
ER	171	8.87E−15	
Positive	93	4.99 ± 1.710			
Negative	80	2.75 ± 1.747			
PR	172	4.01E−09	
Positive	72	5.03 ± 1.807			
Negative	102	3.24 ± 1.925			
HER2	24.51	0.002	
Positive	16	5.13 ± 1.258			
Negative	157	3.88 ± 2.098			
Molecular classifications	171	5.69E−18	
TNBC	66	2.41 ± 1.559			
HER2 positive	16	5.13 ± 1.258			
Luminal	90	4.98 ± 1.754			
T stage	170	0.644	
T1	72	4.11 ± 2.140			
T2	89	4.00 ± 2.067			
T3	7	3.14 ± 1.574			
T4	3	4.67 ± 2.309			
N stage	176	0.363	
N0	98	3.76 ± 2.091			
N1	29	4.48 ± 1.573			
N2	40	4.15 ± 2.190			
N3	10	3.80 ± 2.573			
TNM stage	172	0.859	
I	43	3.86 ± 2.145			
II	74	4.05 ± 1.958			
III	56	4.07 ± 2.190			
Note:

P < 0.05 is considered as statistically significant.

Table 2 Basal statistics of the mRNA datasets.

Datasets	TNBC	Non-TNBC	Sources	
N	M	SD	N	M	SD	
GPL13607	10	8.96	0.22	240	9.81	0.91	GSE59246 USA, GSE70951-GPL13607 USA	
GPL1390	47	0.21	0.29	130	0.30	0.31	GSE10885-GPL1390 USA, GSE10886 USA, GSE10893-GPL1390 USA, GSE1992-GPL1390 USA, GSE2607-GPL1390 USA, GSE6128 USA	
GPL17586	175	6.77	0.62	113	7.41	0.60	GSE115144 China, GSE73540 Malaysia, GSE76250 China, GSE134359 Mexico	
GPL570	304	5.64	0.73	1781	5.86	0.70	GSE20711 Canada, GSE45827 France, GSE65194 France, GSE29431 Spain, GSE7904 USA, GSE31448 France, GSE29044 Saudi Arabia, GSE50567 Poland, GSE61304 Singapore, GSE42568 Ireland, GSE5764 Czech Republic, GSE10780 USA, GSE10810 Spain, GSE21422 Germany, GSE22544 USA, GSE25407 USA, GSE26910 Italy, GSE54002 Singapore, GSE71053 Denmark, GSE147472 Italy, GSE140494 Germany, GSE146558 China, GSE103865 China, GSE153796 USA, GSE135565 South Korea, GSE7307 USA, GSE3744 USA	
GPL6244	41	7.56	0.33	263	7.79	0.46	GSE36295 Saudi Arabia, GSE86374 Mexico, GSE61724 Australia, GSE37751 USA, GSE118432 USA, GSE81838 USA,	
GPL6848	20	−0.09	0.36	153	0.17	0.47	GSE26304 Canada, GSE18672 Norway	
GPL8269	57	0.97	0.21	169	1.13	0.24	GSE22384-GPL8269 USA, GSE41119 USA	
GPL887	67	−0.22	0.50	155	−0.07	0.38	GSE10885-GPL887 USA, GSE2607-GPL887 USA, GSE24124 China, GSE9309 China	
GPL96	24	7.54	0.22	407	7.79	0.56	GSE15852 Malaysia, GSE5364 Singapore, GSE6883-GPL96 USA, GSE158309 Germany	
GSE29174	70	0.02	0.26	80	0.39	0.26	GSE29174 USA	
GSE50428	6	7.13	0.23	20	7.03	0.37	GSE50428 Germany	
METABRIC	199	−0.14	1.01	1705	0.02	1.00	METABRIC	
TCGA	115	2.32	0.41	982	2.73	0.47	TCGA	

Figure 1 EZH1 was downregulated in TNBC tissue samples.

Both (A) EZH1 protein and (B) EZH1 mRNA expression levels were decreased in TNBC tissue compared with non-TNBC tissue. (C) EZH1 protein and (D, E) EZH1 mRNA displayed moderate discriminatory ability in differentiating between TNBC and non-TNBC tissue samples (area under the curve >0.7). TNBC, triple-negative breast cancer. *, P < 0.05; **, P < 0.01; ***, P < 0.001; ****, P < 0.0001.

Figure 2 Immunohistochemistry staining of EZH1 in normal breast tissue samples.

The normal ductal epithelium was intact. EZH1 antibody was stained in the nucleus. (A, B) score 2 points (200×, 400×); (C, D) score 4 points (200×, 400×); (E, F) score 8 points (200×, 400×); (G, H) score 12 points (200×, 400×).

Figure 3 Immunohistochemistry staining of EZH1 in triple-negative breast cancer (TNBC).

Pathological mitosis could be detected in TNBC tissue. EZH1 antibody was stained in the nucleus. (A, B) score 0 points (200×, 400×); (C, D) score 2 points (200×, 400×); (E, F) score 4 points (200×, 400×).

Figure 4 Immunohistochemistry staining of EZH1 in non-triple-negative breast cancer (non-TNBC).

Pathological mitosis could be detected in non-TNBC tissue. EZH1 antibody was stained in the nucleus. (A, B) score 0 points (200×, 400×); (C, D) score 4 points (200×, 400×); (E, F) score 8 points (200×, 400×).

In summary, EZH1 was significantly downregulated in TNBC tissue, with a SMD of −0.59 [ −0.80, −0.37], and no obvious publication bias or heterogeneity was identified (Begg’s test: continuity corrected P = 0.669; Egger’s test: P = 0.224) (Figs. S3A–S3F). Additionally, EZH1 indicated moderate discriminatory ability between TNBC and non-TNBC tissue, as is shown in SROC (AUC = 0.73 [0.69, 0.76]) and diagnostic likelihood ratios (positive ratio: 1.77 [1.49, 2.11]; negative ratio: 0.44 [0.35, 0.56]). In contrast, EZH2 was significantly upregulated in TNBC tissue (SMD > 0), without obvious publication bias or heterogeneity (Figs. S4A–S4C). Additionally, EZH2 also had a moderate discriminatory ability between TNBC and non-TNBC tissue (area under the curve >0.7) (Fig. S4D).

CRISPR screening analysis of EZH1/EZH2 in BC cell lines

Figures. S5A and S5B showed the expression levels of EZH1/EZH2 in a total of 62 different BC cell lines. According to the CCLE, the expression trends of EZH1/EZH2 in BC cell lines were consistent with that in BC tissue. As is opposite to EZH2, EZH1 was significantly downregulated in BC (Fig. 5A) or TNBC (Fig. 5B) cells in comparison with non-cancerous cells. Unexpectedly, the expression discrepancy of EZH1/EZH2 in TNBC cells was insignificant when compared with non-TNBC cells, which may attribute to the obvious heterogeneity in different TNBC cell lines (Fig. 5C). The CRISPR screening CERES of EZH1/EZH2 were negative in different BC cell lines, including TNBC cells, thus indicating that EZH1/EZH2 may be important in the proliferation and survival of BC cells (Fig. 5D).

Figure 5 Expression validation and CRISPR screening of EZH1/EZH2 in breast cancer cells.

The expression and potential biological functions of EZH1/EZH2 were validated in breast cancer cells using Cancer Cell Line Encyclopedia and the Dependency Map. (A) EZH1 was significantly downregulated in breast cancer cells in comparison with non-cancerous cells. However, EZH2 was overexpressed in breast cancer cells when compared with non-cancerous cells. (B) EZH1 was significantly downregulated in triple-negative breast cancer cells in comparison with non-cancerous cells. However, EZH2 was overexpressed in triple-negative breast cancer cells when compared with non-cancerous cells. (C) Because of the heterogeneity in different cell lines, the expression discrepancy of EZH1/EZH2 in triple-negative breast cancer cells was insignificant when compared with non-triple-negative breast cancer cells. (D) The CRISPR screening gene effect scores (CERES) of EZH1/EZH2 were less than zero in different breast cancer cells, thus indicating that EZH1/EZH2 may be important in the proliferation and survival of breast cancer cells.

EZH1 downregulation presaged poor prognosis in TNBC patients

The prognostic value of EZH1 in BC has not been reported, so the authors explored the potential prognostic implication of EZH1. As is shown in Figs. 6A–6C, a lower EZH1 expression was significantly correlated with worse OS (P = 9.5 e−5), DMFS (P = 2.3 e−6), and RFS (P = 2.5 e−16) in BC patients (all with HR <1). Moreover, TNBC patients with lower EZH1 expression were likely to exhibit poorer OS (P = 0.1), DMFS (P = 0.054), and PFS (P = 0.0037) (Figs. 6D–6F). In this setting, EZH1 downregulation may presage poor prognosis of TNBC patients.

Figure 6 EZH1 downregulation presaged poor prognosis in TNBC patients.

Lower EZH1 expression was significantly correlated with worse (A) OS, (B) DMFS, and (C) RFS in BC patients. TNBC patients with lower EZH1 expression were likely to exhibit poorer (D) OS, (E) DMFS, and (F) PFS. TNBC, triple-negative breast cancer; OS, overall survival; DMFS, distal metastasis-free survival; PFS, prognosis-free survival.

In contrast, higher EZH2 expression was significantly correlated with worse OS, DMFS, and RFS in BC patients (Figs. S6A–S6C). However, TNBC patients with lower EZH2 expression were likely to exhibit poorer OS, DMFS, and PFS (Figs. S6D–S6F). Therefore, EZH2 upregulation presaged distinct prognosis in BC and TNBC patients.

A prognostic signature model predicted survival probability of TNBC patients

Because TNBC was a heterogeneous malignancy, the authors tried to establish a model to forecast the survival probability of TNBC patients. Based upon KM survival analysis and HR pooling analysis, a total of six potent prognosticators were identified from TNBC patients (Figs. 7A–7F). Among them, AKAP12, GOLGA1, ITGB3, MATN3, and PCDHB5 were the risk factors for TNBC patients (all with HR >1) while FABP7 was a protective factor for TNBC patients (HR <1) (Figs. S7A–S7F). The risk score equation could be expressed as follows, risk score = 0.3577 * AKAP12 + ( −0.1508) * FABP7 + 0.1570 * GOLGA1 + 0.2151 * ITGB3 + 0.2390 * MATN3 + 0.1484 * PCDHB5. Surprisingly, higher risk scores were believed to predict worse OS outcomes of TNBC patients (Fig. 7G). Therefore, a Cox risk model was established using these six prognostic signatures (Fig. 7H), which exhibited an outstanding performance in predicting the prognosis of TNBC patients at 3, 5, and 10 years (all with AUC >0.70) (Fig. 7I). Moreover, the six-signature-based model presented a moderate ability to forecast the prognosis of TNBC patients in the GSE25307 training cohort (AUC at 10 years >0.70) (Fig. 7J).

Figure 7 A prognostic signature model predicted survival probability in TNBC patients.

(A–G) A total of six prognostic signatures were identified from large cohorts of TNBC patients (i.e., TCGA, METABRIC, and GSE25307) by comprehensively conducting Kaplan–Meier survival analysis, univariate Cox regression analysis, and hazard ratio pooling analysis. (H, I) The six-signatures-based model exhibited phenomenal performance in predicting the prognosis of TNBC patients at 3, 5, and 10 years. (J) The six-signatures-based model presented a moderate ability to forecast the prognosis of TNBC patients in the training cohort (i.e., GSE25307). TNBC, triple-negative breast cancer.

Prognostic signatures facilitated risk stratification of TNBC patients

The prognostic ability of the identified signatures was presented above; the authors subsequently explored the classification ability of such prognostic signatures. Delta area plot, cumulative distribution function graph, and heatmap of consensus matrix all signified that when k = 3, the clustering result was relatively stable; therefore, a total of three clusters were identified from TNBC patients (cluster 1, cluster 2, and cluster 3) (Figs. 8A–8C).

Figure 8 Prognostic signatures contributed to risk stratification in TNBC patients.

(A) Delta area plot, (B) cumulative distribution function graph, and (C) heatmap of consensus matrix all signified that when k = 3, the clustering result was relatively stable. Therefore, a total of three clusters were identified from TNBC patients (cluster 1, cluster 2, and cluster 3). (D) The tumor immune microenvironments were compared between different TNBC clusters. Compared with cluster 1, cluster 2–3 had lower infiltration levels of activated CD4+ T cell. (E) EZH1 expression levels were positively correlated with the infiltration degrees of memory B, plasmacytoid dendritic CD4+ T, effector memory CD4+ T, activated B, and natural killer cells in TNBC patients. (F) Cluster 1 TNBC patients showed significant overall survival advantage compared with cluster 2–3 TNBC patients. TNBC, triple-negative breast cancer.

As is shown in Fig. 8D, the tumor immune microenvironments were compared between different TNBC clusters. Intriguingly, when compared with cluster 1, cluster 2–3 had lower infiltration levels of activated CD4+ T cell (P = 0.032). Additionally, there was a positive correlation between EZH1 expression and the infiltration levels of natural killer (P = 0.0448), activated B (P = 0.0317), effector memory CD4+ T (P = 0.0051), plasmacytoid dendritic cells (P = 0.0031), and memory B cells (P = 0.0003) (all with correlation coefficients >0) (Fig. 8E). Nonetheless, a negative correlation was detected between EZH1 expression values and the infiltration levels of neutrophils, CD56bright natural killer, and type 17 T helper (all with correlation coefficients <0, and P < 0.05). More importantly, subtype 1 TNBC patients displayed significant survival advantage when compared to subtype 2–3 TNBC patients (P = 0.029) (Fig. 8F). In this context, the identified prognostic signatures may be helpful for identifying a malignant TNBC subtype, which was characterized by low activated CD4+ T cell infiltration and poor prognosis.

EZH1 and EZH2 somatic mutations displayed mutual exclusivity in breast cancer tissue

The expression status and prognostic values of EZH1 and EZH2 had been elucidated, and the authors next explored their mutation landscapes in BC and TNBC. As is shown in the Fig. 9A, amplification was the most common mutation type for EZH1 and EZH2 in BC tissue. It was noteworthy that there was a tendency of mutual exclusivity between EZH1 and EZH2 mutations, indicating that EZH1 and EZH2 may play a complementary role for their transcriptional repression in BC. The authors further compared the somatic-mutation types of different TNBC subtypes, and noticed that subtype 2–3 TNBC displayed a higher mutation frequency than subtype 1 TNBC. Additionally, TP53 was the most frequently mutated gene in both subtype 2–3 TNBC and subtype 1 TNBC. For subtype 1 TNBC patients, there was significant co-occurrence between TP53 mutations and tumor suppressor gene PTEN mutations (Fig. 9B). However, TP53 mutations co-occurred with PIK3CA mutations in subtype 2–3 TNBC patients (Fig. 9C). Therefore, the genetic mutations of TNBC were heterogeneous, and a high mutation level of subtype 2–3 TNBC may be responsible for its poor prognosis, compared with subtype 1 TNBC.

Figure 9 EZH1 and EZH2 somatic mutations displayed mutual exclusivity in breast cancer tissue.

(A) Amplification was the most common mutation type of EZH1 and EZH2 in breast cancer tissue. There was a tendency of mutual exclusivity between EZH1 and EZH2 mutations. (B) For subtype 1 TNBC patients, TP53 was the most frequently mutated gene and co-occurred with tumor suppressor gene PTEN mutation. (C) TP53 was the most frequently mutated gene and co-occurred with PIK3CA mutation in subtype 2–3 TNBC patients. TNBC, triple-negative breast cancer. *, P < 0.05.

EZH1 may mediate competing endogenous RNA mechanisms in TNBC

Subsequently, the authors explored the potential EZH1-miRNA-lncRNA regulatory mechanisms in TNBC. We first identified 61 DEmiRNAs and 156 DElncRNAs from TNBC tissue compared with non-TNBC tissue (Figs. 10A and 10B). After intersecting by the predicted miRNA targets of EZH1, hsa-miR-17-5p, hsa-miR-20a-5p, and hsa-miR-942-5p were determined to be potential miRNA targets of EZH1 (Fig. 10C), all of which were significantly upregulated in TNBC. Then the DElncRNAs were intersected by the binding targets of such three miRNAs, and BLACAT1, LINC00839, and XIST were identified as the putative DEmiRNAs targets. Finally, a potential competing endogenous RNA regulatory network of EZH1 in TNBC was established (Fig. 10D), and the binding sites for EZH1 and hsa-miR-17-5p, hsa-miR-20a-5p, and hsa-miR-942-5p were presented in Fig. 10E.

Figure 10 EZH1 may mediate competing endogenous RNA networks in TNBC.

Based on The Cancer Genome Atlas TNBC dataset, (A) DEmiRNAs and (B) DElncRNAs were identified from TNBC patients compared with non-TNBC patients (log2|foldchange| > 1, P < 0.05). (C) Targeted miRNAs that may bind to EZH1 were predicted by intersecting DEmiRNAs and putative EZH1-targeted miRNAs from miRTarBase and TarBase V.8. (D) The potential EZH1-miRNAs-lncRNAs regulatory axis in TNBC was presented. (E) The putative bind sites of EZH1 and hsa-miR-17-5p, hsa-miR-20a-5p, and hsa-miR-942-5p in TNBC were displayed. DEmiRNA, differentially expressed microRNA; DElncRNAs, differentially expressed long non-coding RNAs; TNBC, triple-negative breast cancer.

EZH1 downregulation may promote cell cycle signaling pathway

The molecular mechanisms of EZH1 in TNBC were then investigated. This study identified a total of 8,113 TNBC DEGs, containing 4,215 upregulated DEGs and 3,898 downregulated DEGs in TNBC tissue. Since EZH1 participated in transcriptional repression of its target genes, EZH1 downregulation was anticipated to activate its transcriptional targets. Therefore, we intersected the upregulated TNBC DEGs, EZH1 negative CEGs, and putative EZH1 targets, where a total of 789 targets were identified (Fig. 11A). We clustered the GO terms by similarity and found that the upregulated DEGs and transcriptional target genes were obviously aggregated in the cell-cycle-associated biological process and molecular functions (Fig. 11B). In addition, cell cycle was detected to be the most enriched KEGG pathway (Fig. 11C). More interestingly, such upregulated DEGs and transcriptional targets were mapped for BC disease (Fig. 11D). In all, EZH1 downregulation may activate cell cycle signaling pathway, where CHEK1 was identified as the hub gene (Fig. 11E).

Figure 11 EZH1 downregulation may activate cell cycle pathway in TNBC.

(A) EZH1-negative CEGs were intersected by upregulated genes and the predicted transcriptional targets of EZH1 in TNBC. (B) Mitotic cell cycle regulation was the most enriched gene ontology term. (C) Cell cycle was identified as the most enriched Kyoto Encyclopedia of Genes and Genomes pathway in TNBC. (D) Hereditary breast ovarian cancer was identified as the most clustered disease. (E) Putative transcriptional factor binding sites for EZH1 and CCNA2, CCNB1, MAD2L1, and PKMYT1, all of which were identified from the cell-cycle pathway. CEGs, co-expressed genes; TNBC, triple-negative breast cancer.

EZH2 upregulation may suppress the response to hormone in TNBC tissue

Since EZH2 was involved in the transcriptional repression of its targets, we intersected the putative EZH2 targets, downregulated TNBC differentially expressed genes, and EZH2 negative co-expressed genes, where a total of 770 targets were identified (Fig. S8A). The downregulated EZH2 transcriptional targets were enriched in response to the hormone GO term and pathways in cancer (Fig. S8B). Furthermore, ESR1 (estrogen receptor 1) was identified as a key gene in response to the hormone (Fig. S8C). The transcriptional factor binding sites for EZH2 and ESR1 was predicted (Fig. S8D).

EZH1 may transcriptionally suppress cell-cycle-associated genes

Genes mapped to the cell-cycle KEGG pathway were selected for further analyses. The authors have elaborated that EZH1 and its transcriptional targets exhibited converse expression trend in TNBC. Additionally, according to the Pearson correlation analysis, a negative correlation was found between the mRNA expression levels of EZH1 and CCNA2 (cyclin A2), CCNB1 (cyclin B1), MAD2L1 (mitotic arrest deficient 2 like 1), and PKMYT1 (protein kinase, membrane associated tyrosine/threonine 1) (Table S3). Based on the public ChIP-seq data, the putative transcriptional factor binding sites for EZH1 and CCNA2, CCNB1, MAD2L1, and PKMYT1 were identified (Fig. 11F). In this setting, it was reasonable to propose that EZH1 might transcriptionally repress such cell-cycle-associated genes. When EZH1 was downregulated in TNBC, cell cycle genes, CCNA2, CCNB1, MAD2L1, and PKMYT1, may be activated. More intriguingly, such potential EZH1 targets were negatively correlated to the immune infiltration levels in TNBC tissue (Fig. S9).

Parthenolide was identified as a potential therapeutic agent for treating TNBC patients

Finally, the authors explored the novel therapeutic small molecules for TNBC patients by targeting cell cycle pathway, and the top four small molecules (i.e., parthenolide, emodic acid, cyclosporin-a, and naproxol) for treating TNBC are predicted in Table 3. Parthenolide molecules were used to dock with the transcriptional targets of EZH1 (i.e., CCNA2, CCNB1, MAD2L1, and PKMYT1) (Fig. 12). The binding energy for parthenolide ligand and CCNB1 protein was the lowest, indicating that CCNB1 seems most effective in conjunction with parthenolide. Moreover, it was observed that parthenolide formed hydrogen bond with the 300th amino acid of CCNB1 protein, leucine, which also implied the stability of the docking result.

Table 3 Repurposing drugs by targeting the cell-cycle-and FoxO-signaling pathway in triple-negative breast cancer (TNBC).

Perturbation id	Perturbations	Cell line	Dose	Time	Samples	Mechanism of action	Raw connectivity scores	Normalized connectivity score	
BRD-K98548675	Parthenolide	SKMEL5	10 μM	24 h	3	NFKB inhibitor	−0.81	−1.98	
BRD-K94841585	Emodic-acid	MCF7	10 μM	24 h	4	Laxative	−0.8	−1.95	
BRD-K03222093	Cyclosporin-a	MCF7	10 μM	24 h	3	/	−0.8	−1.94	
BRD-K34014345	Naproxol	MCF7	10 μM	24 h	5	Anti-inflammatory	−0.8	−1.94	

Figure 12 Parthenolide ligand was docked by the transcriptional targets of EZH1.

Parthenolide was docked by the EZH1 targets as follows, (A) CCNA2. (B) CCNB1. (C) MAD2L1. (D) PKMYT1. The binding energy for parthenolide ligand and CCNB1 protein was the lowest, indicating that CCNB1 seems most effective in conjunction with parthenolide.

Discussion

TNBC constitutes the most malignant BC subtype, which is characterized by high heterogeneity, invasiveness, and poor prognosis (Kajihara et al., 2020; Morotti et al., 2021). The prognostic stratification and personalized therapeutics of TNBC patients are urgently needed. EZH1 has been demonstrated to play a key role in the development of various cancers (Wassef et al., 2019), such as multiple myeloma (Nakagawa et al., 2019), lymphoma (Kagiyama et al., 2021), and thyroid tumors (Jung et al., 2018). Nonetheless, the expression patterns of EZH1 in BC remain controversial. This study not only verified the opposite expression patterns of EZH1 and EZH2, but also revealed their distinct mutation landscapes and the potential transcriptional mechanisms in TNBC. EZH1 downregulation was believed to promote cell cycle pathway by transcriptionally regulating the CCNB1 target. Nonetheless, EZH2 was significantly upregulated and may suppress response to the hormone by regulating ESR1. EZH1 and EZH2 somatic mutations may be mutually exclusive in BC tissue. Finally, parthenolide was predicted to be a potential small molecule agent in treating TNBC by targeting CCNB1.

This is the first multi-centered study implicating EZH1 in TNBC, and we comprehensively compared the expressional and mutational discrepancy of EZH1/EZH2. In-house TMAs (66 TNBC vs. 106 non-TNBC) and external gene microarrays, as well as RNA-seq datasets (1,135 TNBC vs. 6,198 non-TNBC), were utilized to confirm the downregulation of EZH1 at both the protein and mRNA levels. Previous studies have drawn different conclusions on the expression levels of EZH1 in BC tissue, which may contribute to a small sample size (Liu et al., 2012; Zeng, Yang & Wu, 2019). According to the existing literature, a larger sample size helps to ensure enhanced precision and meaningful comparison (Maggio & Franklin, 2020). In this setting, the downregulation of EZH1 in TNBC compared with non-TNBC patients was more significant, which correlated with poorer prognosis in TNBC patients. Opposite to EZH1, EZH2 was significantly upregulated in TNBC. More intriguingly, although EZH2 upregulation presaged poor survival outcome in BC patients, its overexpression may be a protective factor in TNBC patients (HR < 0), which seemed to be contradictory to the former study (Yomtoubian et al., 2020). It has been reported that higher EZH2 expression may lead to worse OS in 2,330 BC patients (Wang et al., 2015). Moreover, EZH2 expression was observed to promote the proliferation, migration, and invasion abilities of TNBC cells (Gao et al., 2020). Nonetheless, our study may elucidate the distinct prognostic role of EZH2 in BC or TNBC patients. Besides, there seemed to be obvious mutual exclusiveness between the mutations of EZH1/EZH2, which may be consistent with their roles as two mutually exclusive catalytic subunits of PRC2 (Lee et al., 2018).

Furthermore, this study preliminarily compared the transcriptional regulation mechanisms of EZH1/EZH2 in TNBC using a method of in-silico analysis. Accumulating studies have demonstrated the tumorigenic function of EZH2 (Honma et al., 2017; Hsu et al., 2020), but the roles of EZH1 in carcinogenesis were rarely known. Herein, the transcriptional mechanisms of EZH1 in the cell-cycle pathway were first explored. As in known, EZH1 has been demonstrated to be positively relevant to active transcription on a genome-wide scale (Su et al., 2016). For example, EZH1 and SUZ12 PRC2 subunits (SUZ12) formed PRC2 and positively activated the expression levels of erythroid-differentiation-related genes in human erythroid precursors (Xu et al., 2015). In TNBC, it is noteworthy that the upregulated putative targets of EZH1 were significantly aggregated in the cell cycle molecular pathway, where a total of four genes (i.e., CCNA2, CCNB1, MAD2L1, and PKMYT1) were determined as key transcriptional targets of EZH1. Intriguingly, EZH1 itself is essential for preventing senescence-like cell-cycle arrest (Hidalgo et al., 2012), indicating its positive roles in cell-cycle progression. CCNA2 and CCNB1 are both cyclin members and promote cell cycle progression by interacting with cyclin-dependent kinase (CDK) (Silva Cascales et al., 2021); however, CDK could be inactivated by PKMYT1 through phosphorylation, thus regulating the cell cycle (Lee & Yang, 2001). Additionally, spindle assembly checkpoint gene MAD2L1 participates in cell cycle mitosis and it can effectively inhibit the onset of anaphase phase until all chromosomes have been properly aligned (Zhang & Nilsson, 2018). It is evidenced that EZH1 was associated with the silenced CCNA2 gene, indicating a negative transcriptional regulation between EZH1 and CCNA2 (Cheedipudi et al., 2015), which was consistent with the opposite expression trend of decreased EZH1 and increased CCNA2 in TNBC tissue. Although the potential relevance between EZH1 and the other three targets has not been revealed, their upregulated expression levels and pro-tumor roles have been elucidated in BC or even TNBC (Ben-Hamo et al., 2020; Deng et al., 2020; Hong et al., 2021; Liu et al., 2020; Qi et al., 2019). Collectively, it was conceivable to propose that EZH1 may induce the malignant behaviors of TNBC by transcriptionally regulating cell-cycle-enriched genes. In the future, the transcriptional regulatory mechanisms of EZH1 in TNBC cells deserves further validation.

Regarding the downregulated putative targets of EZH2 in TNBC, response to hormone was the most enriched biological function. As is well-known, EZH2 has been a hotspot of cancerous research (Duan, Du & Guo, 2020; Li et al., 2020; Wen et al., 2021). In TNBC cells, EZH2 could promote the migration and invasion abilities by the transcriptional repression of tissue inhibitor of metalloproteinase 2 and the activation of matrix metalloproteinase 2 (MMP-2) and MMP-9 (Chien et al., 2018). In the present study, we shed light on the transcriptional repression EZH2 had on response to the hormone, where ESR1 was identified as the hub transcriptional target. We observed that upregulated EZH2 and downregulated ESR1 was negatively correlated in TNBC. Fortunately, the intimate association between EZH2 and ESR has been previously described. In a TNBC mice model, EZH2 and CDK2 inhibition may induce the re-expression of estrogen receptor α (ERα), and more amazingly, may even convert TNBC into luminal ERα-positive BC (Nie et al., 2019). Additionally, in ovarian carcinoma, reduced EZH2 phosphorylation caused by CDK2 inhibition was found to activate downstream ESR1 (Han et al., 2020). Taken together, we proposed that upregulated EZH2 may transcriptionally repress response to the hormone by downregulating ESR1. However, further experimental verifications must be provided in future studies.

More importantly, this study was dedicated to repurposing novel therapeutic molecules for TNBC patients by targeting the prospective transcriptional mechanisms of EZH1 in the cell-cycle pathway. Among the predicted small molecules, parthenolide, emodic acid, cyclosporine-a, and naproxol have been experimentally elucidated in treating BC. Moreover, it was inferred that CCNB1 seems most effective in conjunction with parthenolide. Fortunately, the inhibitory effect that parthenolide has on TNBC cells has been confirmed previously (Araújo et al., 2020; Ge et al., 2019; Ghorbani-Abdi-Saedabad et al., 2020; Jin et al., 2020; Sufian et al., 2022). It was reported that parthenolide could attenuate the proliferation and survival of BC cells by covalently modifying focal adhesion kinase (Berdan et al., 2019). Additionally, DMOCPTL, a derivative of parthenolide, showed obvious anti-TNBC effect by inducing ferroptosis mechanisms (Ding et al., 2021). Therefore, parthenolide and its derivatives might be promising in treating TNBC patients in the future. According to the previous studies, pathenolide could exhibit an anti-inflammation activity in several autoimmune diseases (Jones et al., 2021; Wang et al., 2016; Zhang et al., 2022). In another study, it was noticed that pathenolide exerted protective function in the apoptosis of peripheral blood T cells (Li-Weber et al., 2002). The authors were interested in the effect of pathenolide on immune cells in TNBC tissue. Surprisingly, a negative correlation was detected between the expression levels of CCNB1—a prospective target of pathenolide—and the immune infiltration levels in TNBC tissue. In this context, pathenolide is likely to improve the anti-tumor immunity in TNBC tissue and promote the apoptosis of TNBC cells, which requires in-depth validation. Also, emodic acid exhibited efficacy in inhibiting the migration and invasiveness of 4T1 BC cells (Abdellatef et al., 2021). Furthermore, cyclosporine-a was demonstrated to display an anti-tumor effect and has been designed as a potential medical device for the treatment of BC patients (Trombino et al., 2021). Last, naproxol—a treatment for cancer-related pain(Kirshner et al., 2018)—also displayed an anti-cancer effect (Deb et al., 2014), which could have been induced by early apoptosis. Taken together, this study provides an avenue for the treatment prospects of TNBC patients by targeting the transcriptional network of EZH1. Nevertheless, more studies must be conducted to put this into practice.

Finally, the six-signature-based prognostic model produced an impressive performance in this study. By quantitatively analyzing the expression levels of AKAP12, FABP7, GOLGA1, ITGB3, MATN3, and PCDHB5, the established prognostic model could be utilized to forecast the survival probability of TNBC patients, with a training AUC of 0.753, 0.981, and 0.977 at 3, 5, and 10 years, respectively. Furthermore, the validating AUC displayed moderate sensitivity (AUC = 0.708) in predicting the 10-year survival rate of TNBC patients. However, the practical efficacy of the prognostic model in the risk stratification of TNBC patients must be certified in future clinical studies.

There were serval limitations in this study. The biological functions of EZH1 in TNBC cells have not been explored by using in vitro and in vivo experiments. Additionally, the effect of parthenolide have not been analyzed readily in TNBC cell culture models. However, the present study comprehensively analyzed the in-silico results and CRISPR screening data, as well as scientific literatures results. Our study revealed the potential role of EZH1 in promoting TNBC cell growth and explored the therapeutic targets of parthenolide in treating TNBC cells. More experimental evidence must be supplemented in the future study.

Conclusions

EZH1 was significantly downregulated in TNBC tissue when compared with any other BC subtypes. EZH1 could presage poor prognosis in TNBC patients and may be a key modulator in the progression of TNBC by targeting the cell cycle pathway. Parthenolide was predicted to be a promising molecule for treating TNBC, where CCNB1 might be a potential therapeutic target.

Supplemental Information

Supplemental Information 1 Score of EZH1 immunohistochemical staining in each tissue.

Footnote: We used nucleus staining as an indicator of counting. Under a high-powered field of view (10 * 40), we continuously counted five fields, observed the proportion of positive cells and the intensity of nuclear staining in each field, and scored accordingly. (1) The proportion of positive cells (P) is scored as follows: zero point for 0% to 5%, one point for 6% to 25%, two points for 26% to 50%, three points for 51% to 75%, and four points for ≥76%. (2) The intensity (I) of cell nucleus staining is scored on a scale of 0 points for no cell staining, one point for mild cell staining, two points for moderate cell staining, and three points for strong positive staining. The final staining score is calculated by the following algorithm, staining score = P * I. Therefore, the lowest score is zero point, and the highest score is 12 points.

Click here for additional data file.

Supplemental Information 2 The raw data of in-house immunohistochemistry result.

Click here for additional data file.

Supplemental Information 3 A negative correlation was found between EZH1 expression and CCNA2, CCNB1, MAD2L1, and PKMYT1 expression.

Click here for additional data file.

Supplemental Information 4 HE staining of BC and normal breast tissues (200×).

(A, B) HE staining of BC tissues. (C, D) HE staining of normal breast tissues. HE, hematoxylin and eosin. BC, breast cancer.

Click here for additional data file.

Supplemental Information 5 Protein expression levels of EZH1 in different BC subgroups.

The protein expression of EZH1 in normal breast tissue was higher than in (A) BC, (B) TNBC, and (C) non-TNBC tissues. The protein expression of EZH1 in TNBC tissues was lower than in (D) HER2 + BC and (E) luminal BC tissues. (F) There was no significant difference in the protein expression of EZH1 between HER2 + BC and luminal BC tissues. EZH1 protein had a strong ability to differentiate (G) BC, (H) TNBC, and (I) non-TNBC tissues from normal breast tissues. BC, breast cancer; TNBC, triple-negative breast cancer.

Click here for additional data file.

Supplemental Information 6 EZH1 mRNA was lowly expressed in TNBC tissues.

(A) EZH1 was significantly downregulated in TNBC tissues (SMD <0), and no obvious (B) heterogeneity or (C) publication bias was identified (Begg’s test: continuity corrected P = 0.669; Egger’s test: P = 0.224). (D) EZH1 had a moderate discriminatory ability between TNBC and non-TNBC tissues (area under the curve >0.7). The overall accuracy of EZH1 in discriminating TNBC from non-TNBC tissues were explored using (E) positive likelihood ratio forest plot and (F) negative likelihood ratio forest plot.

Click here for additional data file.

Supplemental Information 7 EZH2 mRNA was overexpressed in TNBC tissues.

Unlike EZH1, (A) EZH2 was significantly upregulated in TNBC tissues (SMD >0), without obvious (B) heterogeneity or (C) publication bias (Begg’s test: continuity corrected P = 0.127; Egger’s test: P = 0.053). (D) EZH2 had a moderate discriminatory ability between TNBC and non-TNBC tissues (area under the curve >0.7). TNBC, triple-negative breast cancer.

Click here for additional data file.

Supplemental Information 8 EZH1/EZH2 expression levels in breast cancer cell line.

EZH1/EZH2 expression were validated in breast cancer cell lines using Cancer Cell Line Encyclopedia. (A) EZH1 (B) EZH2.

Click here for additional data file.

Supplemental Information 9 EZH2 upregulation presaged distinct prognosis in BC and TNBC patients.

Higher EZH2 expression was significantly correlated with worse (A) OS, (B) DMFS, and (C) RFS in BC patients. TNBC patients with lower EZH2 expression were likely to exhibit poorer (D) OS, (E) DMFS, and (F) PFS. TNBC, triple-negative breast cancer; OS, overall survival; DMFS, distal metastasis-free survival; PFS, prognosis-free survival.

Click here for additional data file.

Supplemental Information 10 Hazard ratio values were pooled for prognostic signatures in triple-negative breast cancer.

A: AKAP12. B: FABP7. C: GOLGA1. D: ITGB3. E: MATN3. F: PCDHB5.

Click here for additional data file.

Supplemental Information 11 Upregulated EZH2 may suppress the response to hormone in TNBC tissues.

(A) Since EZH2 was involved in the transcriptional repression of its targets, we intersected the putative EZH2 targets, downregulated TNBC differentially expressed genes, and EZH2 negative co-expressed genes, where a total of 770 targets were identified. (B) The downregulated EZH2 transcriptional targets were enriched in response to hormone. (C) ESR1 was identified as a key gene in response to hormone. (D) The transcriptional factor binding sites for EZH2 and ESR1 was predicted. TNBC, triple-negative breast cancer.

Click here for additional data file.

Supplemental Information 12 Negative correlation was observed between the potential EZH1 targets and immune infiltrations in triple-negative breast cancer.

CCNB1 expression level was negatively correlated to the infiltration levels of central memory CD4+ T cell, type 17 T helper cell, and T follicular helper cell. However, a significantly positive correlation could be found between the expression level of CCNB1 and the infiltration levels of activated CD4+ T cell and type 2 T helper cell.

Click here for additional data file.

We are grateful for the technological support from Guangxi Key Laboratory of Medical Pathology.

Additional Information and Declarations

Competing Interests

Author Contributions

Data Availability

The authors declare that they have no competing interests.

Wei Peng conceived and designed the experiments, analyzed the data, prepared figures and/or tables, authored or reviewed drafts of the article, and approved the final draft.

Wei Tang conceived and designed the experiments, authored or reviewed drafts of the article, and approved the final draft.

Jian-Di Li conceived and designed the experiments, analyzed the data, prepared figures and/or tables, authored or reviewed drafts of the article, and approved the final draft.

Rong-Quan He performed the experiments, authored or reviewed drafts of the article, and approved the final draft.

Jia-Yuan Luo performed the experiments, analyzed the data, authored or reviewed drafts of the article, and approved the final draft.

Zu-Xuan Chen performed the experiments, authored or reviewed drafts of the article, and approved the final draft.

Jiang-Hui Zeng performed the experiments, authored or reviewed drafts of the article, and approved the final draft.

Xiao-Hua Hu performed the experiments, authored or reviewed drafts of the article, and approved the final draft.

Jin-Cai Zhong performed the experiments, authored or reviewed drafts of the article, and approved the final draft.

Yang Li performed the experiments, authored or reviewed drafts of the article, and approved the final draft.

Fu-Chao Ma performed the experiments, authored or reviewed drafts of the article, and approved the final draft.

Tian-Yi Xie performed the experiments, authored or reviewed drafts of the article, and approved the final draft.

Su-Ning Huang conceived and designed the experiments, authored or reviewed drafts of the article, and approved the final draft.

Lian-Ying Ge conceived and designed the experiments, authored or reviewed drafts of the article, and approved the final draft.

The following information was supplied regarding data availability:

The raw data is available in the Supplemental Files.

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
