# Peer review of "Downregulation of the enhancer of zeste homolog 1 transcriptional factor predicts poor prognosis of triple-negative breast cancer patients"

_PeerJ, doi:10.7717/peerj.13708_

## Round 0.1 · original submission · Major Revisions

All reviewers agree that this is an interesting and worthy study. I concur. However, I am bothered by the lack of experimental biology to validate some of this data, e.g. I am surprised you have not performed some simple in vitro experiments. The effect of inhibitors of these molecules could have been analyzed readily in cell culture models and I am minded to ask for this type of data.

I recognise this is extra work, but this is my preferred option.

Alternatively, you should introduce into your discussion a section labelled 'limitations of the study' and comment on the issues this raises. I believe this must be clearly signposted. However, I stress my preference is for some experimental data.

Please address all the other comments, provide a detailed rebuttal letter explaining your changes and a track-changes version of the paper so I can inspect the changes made.

I look forward to the revised paper.

Reviewer 1 ·

Basic reporting

In this paper, Peng Wei et al. investigate the expression and transcriptional mechanisms of EZH1/EZH2 in triple negative breast cancer tissues. In addition, it has been analyzed the molecular signature as a prognostic value for risk stratification in patients affect by TNBC.

This work is well written and the english language is sufficient quality.
The figures and tables are enough clear. The citations cover the relevant literature adequately.

Experimental design

The methods using in this work are well described.


The statistical method seems valid and correct.

Validity of the findings

The conclusions are very little described (only two lines). We could speculate the role of CCNB1 on immune system cells such as T and NK cells. It has been well described the role of these cells in tumor immunesuveillance. It would also be intersting to investigate the effect of pathenolide on Immune cells.
The impact on and the novelty of this potential therapeutic approach is not well described.

Reviewer 2 ·

Basic reporting

The authors provide novel informaton of the potential role of EZH1 and EZH2 expression in breast cancer. The disruption of the expression of these molecules have been well documented in haematological diseases while less in solid tumors. Moreover, the novelty of the paper is to suggest that while EZH1 upregulation wold seem to favor haematological diseases, in the present manuscript is the downregulaiton of EZH1 that seems to correlate with a poor prognosis of TNBC.
It also of interest the opposite result obtained by analysis of EZH2 , suggesting a non reduntant but rather complementary contribution of the two molecules to the disease .

Experimental design

The reviewer has a little expertise in in silico and clinical analyses , however data are produced by analyzing a high number of samples, and the diffferent type of statistical evaluations performed seem to be coherent with the conclusions. However, it would have been important to validate some of this data performing in vitro experiments . The effect of inhibitors of these moleculse could have been analyze on tumor BC cells( BC cell lines?) showing no perturbations in EZH1 and EZH2 expression and then verify whether these cells would acquire a phenotype similar to that observed in TBNC cells and whether the expression of genes considered target of EZH1 or EZH2 activity resulted modified upon treatment wiht EZH1/2 inhibitors.

Validity of the findings

Again , data appear of great interest even if some in vitro analyses would have help to improve their validity.

Additional comments

NO further comment

Reviewer 3 ·

Basic reporting

Review
Downregulation of the enhancer of zeste homolog 1 transcriptional factor predicts poor prognosis of triple-negative breast cancer patients
The authors in this paper wanted to prove EZH-1 downregulation as a prognostic factor for risk stratification in patients with TNBC. They have used multiple molecular biology and bioinformatics tools, to prove the same. In addition to showing the downregulation of EZH1, they have also discovered downstream upregulated targets and pathways. Not only that, they have even done high throughput screening to identify the small molecule, Parthenolide, to sequester one of the up-regulated proteins, CCNB1, thus proving it as a potential therapeutic agent for TNBC patients. The research that went into this manuscript is commendable. However, I would like to draw the authors' attention to the following discrepancies.
Use colon (:) instead of a full stop at the end of subheadings.
Line 43: Please correct lack to lacks.
Line 58: Correct 66 TN to 66 TNBC.
Line 97: Correct “constructs” to “is”.
Line 111: Add ‘to be” between proved and upregulated.
Line 135: Change were to was.
Line 138: “if they met any standard, as follows” - can be removed from the sentence.
Line 140: Please explain what is a special type of BC.
Line 154: The word “queried” might not be required, as it is implied already.
Lines 289 and 294: Suggest verifying if publication bias can be used in this context.
Line 418: Introduce “the” before response and hormone. Correct mutual to mutually.
Line 470: Correct researches to research.
Line 481: Correct transcriptional to transcriptionally.
Line 504: Correct stratifications to stratification.

General: Figure legends that represent combined results such as (A, B), (C–E), etc., should have proper labeling under the images and/or should be separately notated under a general heading, with a separate mention. For Ex. Instead of the current legend, 1. (C–E) TNBC and non-TNBC tissues (area under the curve > 0.7). TNBC, triple-negative breast cancer, can be modified as EZH1 displayed moderate discriminatory ability in differentiating between D) TNBC and E) non-TNBC.

Experimental design

Introduction
Line 81 - 83: What is the stage of TNBC in non-Hispanic black women that is compared to Stage IV TNBC in Asian women?
Line 111: Correct miRNA193 to miRNA93.

Results and Figures:
EZH1 was downregulated in TNBC tissues:
1. The text states that EZH1 is downregulated, while the legend says the opposite. Correct accordingly. Also, ad space between non-TNBC and TNBC labels on the y-axis.
2. Please mention Fig 1C in the text. It will be helpful if the images are labeled below the graphs to specify the tissue samples (TNBC or non-TNBC), from which the observations are made.
3. Correct line 81 (Fig 1D-1F) to (Fig 1C-1E).
4. Is it possible to put the figures 2 - 4 together under one figure for easy comparison and rearrange other important findings made in this manuscript?
Line 303: EZH1 may be a risk factor for poor prognosis of TNBC patients - EZH1 downregulation worsens prognosis, hence it is not a risk factor.
Figure 7D is encroaching on to Fig 7E. Can it be resized or reorganized?
Label below images 8B, C, and 8D, E as subtype 1 TNBC and subtype 2–3 TNBC, respectively.
Label 8 E, F, and G for their respective miR’s.
Parthenolide was identified as a potential therapeutic agent for treating TNBC patients: This is an important addition to the research, and I would recommend moving the Supplemental figure S8 to the main manuscript.

Validity of the findings

No Comment

Additional comments

No Comment

Annotated reviews are not available for download in order to protect the identity of reviewers who chose to remain anonymous.

Reviewer 4 ·

Basic reporting

This is a well-written article that described a down-regulation of the enhancer of zeste homolog 1 transcriptional factor, with the potential of predicting poor prognosis, in triple-negative breast cancer (TNBC) patients.

TNBC has been recognized as the most malignant subtype of breast cancer and lack effective biomarkers. In addition to reveal the prognostic molecular signatures, this study was designed to unravel the expression status and the prospective transcriptional mechanisms of EZH1/EZH2 in TNBC tissues.

Experimental design

The experiments were designed with target search from previously published and deposited datasets, tissue immunohistochemistry (IHC), applying a prognostic signature model to predict survival and risk probabilities in TNBC patients, determination of somatic mutations at EZH1/EZH2 loci, and prediction of down-regulation of EZH1 in activating cell cycle pathway.

Validity of the findings

This is a bioinformatic article. It was written based on reanalyses of existing data, with lacking of additional wet-lab studies. It is suggested that authors may conduct RT-qPCR to demonstrate the differential expression of EZH1/EZH2 and perform functional analysis to confirm the pathogenic impact of EZH1 on pathway(s).

---

## Round 0.2 · accepted · Accept

Thank you for carefully attending to the reviewers comments. While the review enclosed suggests splitting the paper into two, I strong suggest you do not do this and rather I have accepted the paper as it stands.

Reviewer 3 ·

Basic reporting

Re-review
Downregulation of the enhancer of zeste homolog 1 transcriptional factor predicts poor prognosis of triple-negative breast cancer patients.

The authors in this paper wanted to prove EZH-1 downregulation as a prognostic factor for risk stratification in patients with TNBC. They have used multiple molecular biology and bioinformatics tools, to prove the same. In addition to showing the downregulation of EZH1, they have also discovered downstream upregulated targets and pathways. Not only that, they have even done high throughput screening to identify the small molecule, Parthenolide, to sequester one of the up-regulated proteins, CCNB1, thus proving it as a potential therapeutic agent for TNBC patients. The research that went into this manuscript is commendable.

The authors excellently addressed all the reviewers’ suggestions. In my opinion, the work that went into this paper is expansive. If all other reviewers and editors are satisfied with the additions, I suggest two publications instead of one. If there is a shortage of information for the second publication, they can easily add the supplementary figures as the main figures.

Experimental design

No Comment

Validity of the findings

No Comment